# Unified Language Model Pre-training for Natural Language Understanding and Generation

**Li Dong**[*]  **Nan Yang**[*]  **Wenhui Wang**[*]  **Furu Wei**[*†]  **Xiaodong Liu**  **Yu Wang**
**Jianfeng Gao**  **Ming Zhou**  **Hsiao-Wuen Hon**
Microsoft Research
{lidong1,nanya,wenwan,fuwei}@microsoft.com
{xiaodl,yuwan,jfgao,mingzhou,hon}@microsoft.com

## Abstract

This paper presents a new **UNI**fied pre-trained **L**anguage **M**odel (UNILM) that can be fine-tuned for both natural language understanding and generation tasks. The model is pre-trained using three types of language modeling tasks: unidirectional, bidirectional, and sequence-to-sequence prediction. The unified modeling is achieved by employing a shared Transformer network and utilizing specific self-attention masks to control what context the prediction conditions on. UNILM compares favorably with BERT on the GLUE benchmark, and the SQuAD 2.0 and CoQA question answering tasks. Moreover, UNILM achieves new state-of-the-art results on five natural language generation datasets, including improving the CNN/DailyMail abstractive summarization ROUGE-L to **40.51** (2.04 absolute improvement), the Gigaword abstractive summarization ROUGE-L to **35.75** (0.86 absolute improvement), the CoQA generative question answering F1 score to **82.5** (37.1 absolute improvement), the SQuAD question generation BLEU-4 to **22.12** (3.75 absolute improvement), and the DSTC7 document-grounded dialog response generation NIST-4 to **2.67** (human performance is 2.65). The code and pre-trained models are available at `https://github.com/microsoft/unilm`.

## 1   Introduction

Language model (LM) pre-training has substantially advanced the state of the art across a variety of natural language processing tasks [8, 29, 19, 31, 9, 1]. Pre-trained LMs learn contextualized text representations by predicting words based on their context using large amounts of text data, and can be fine-tuned to adapt to downstream tasks.

Different prediction tasks and training objectives have been used for pre-training LMs of different types, as shown in Table 1. ELMo [29] learns two unidirectional LMs: a forward LM reads the text from left to right, and a backward LM encodes the text from right to left. GPT [31] uses a left-to-right Transformer [42] to predict a text sequence word-by-word. In contrast, BERT [9] employs a bidirectional Transformer encoder to fuse both the left and right context to predict the masked words. Although BERT significantly improves the performance of a wide range of natural language understanding tasks [9], its bidirectionality nature makes it difficult to be applied to natural language generation tasks [43].

In this work we propose a new **UNI**fied pre-trained **L**anguage **M**odel (UNILM) that can be applied to both natural language understanding (NLU) and natural language generation (NLG) tasks. UNILM is a multi-layer Transformer network, jointly pre-trained on large amounts of text, optimized for three types of unsupervised language modeling objectives as shown in Table 2. In particular, we design a

---

[*] Equal contribution. † Contact person.

|  | ELMo | GPT | BERT | UNiLM |
|---|:---:|:---:|:---:|:---:|
| Left-to-Right LM | ✓ | ✓ | | ✓ |
| Right-to-Left LM | ✓ | | | ✓ |
| Bidirectional LM | | | ✓ | ✓ |
| Sequence-to-Sequence LM | | | | ✓ |

Table 1: Comparison between language model (LM) pre-training objectives.

| Backbone Network | LM Objectives of Unified Pre-training | What Unified LM Learns | Example Downstream Tasks |
|---|---|---|---|
| Transformer with shared parameters for all LM objectives | Bidirectional LM | Bidirectional encoding | GLUE benchmark Extractive question answering |
| | Unidirectional LM | Unidirectional decoding | Long text generation |
| | Sequence-to-Sequence LM | Unidirectional decoding conditioned on bidirectional encoding | Abstractive summarization Question generation Generative question answering |

Table 2: The unified LM is jointly pre-trained by multiple language modeling objectives, sharing the same parameters. We fine-tune and evaluate the pre-trained unified LM on various datasets, including both language understanding and generation tasks.

set of cloze tasks [41] where a masked word is predicted based on its context. These cloze tasks differ in how the context is defined. For a left-to-right unidirectional LM, the context of the masked word to be predicted consists of all the words on its left. For a right-to-left unidirectional LM, the context consists of all the words on the right. For a bidirectional LM, the context consists of the words on both the right and the left [9]. For a sequence-to-sequence LM, the context of the to-be-predicted word in the second (target) sequence consists of all the words in the first (source) sequence and the words on the its left in the target sequence.

Similar to BERT, the pre-trained UNiLM can be fine-tuned (with additional task-specific layers if necessary) to adapt to various downstream tasks. But unlike BERT which is used mainly for NLU tasks, UNiLM can be configured, using different self-attention masks (Section 2), to aggregate context for different types of language models, and thus can be used for both NLU and NLG tasks.

The proposed UNiLM has three main advantages. First, the unified pre-training procedure leads to a single Transformer LM that uses the shared parameters and architecture for different types of LMs, alleviating the need of separately training and hosting multiple LMs. Second, the parameter sharing makes the learned text representations more general because they are jointly optimized for different language modeling objectives where context is utilized in different ways, mitigating overfitting to any single LM task. Third, in addition to its application to NLU tasks, the use of UNiLM as a sequence-to-sequence LM (Section 2.3), makes it a natural choice for NLG, such as abstractive summarization and question generation.

Experimental results show that our model, used as a bidirectional encoder, compares favorably with BERT on the GLUE benchmark and two extractive question answering tasks (i.e., SQuAD 2.0 and CoQA). In addition, we demonstrate the effectiveness of UNiLM on five NLG datasets, where it is used as a sequence-to-sequence model, creating new state-of-the-art results on CNN/DailyMail and Gigaword abstractive summarization, SQuAD question generation, CoQA generative question answering, and DSTC7 dialog response generation.

## 2 Unified Language Model Pre-training

Given an input sequence $x = x_1 \cdots x_{|x|}$, UNiLM obtains a contextualized vector representation for each token. As shown in Figure 1, the pre-training optimizes the shared Transformer [42] network with respect to several unsupervised language modeling objectives, namely, unidirectional LM, bidirectional LM, and sequence-to-sequence LM. In order to control the access to the context of the word token to be predicted, we employ different masks for self-attention. In other words, we use masking to control how much context the token should attend to when computing its contextualized

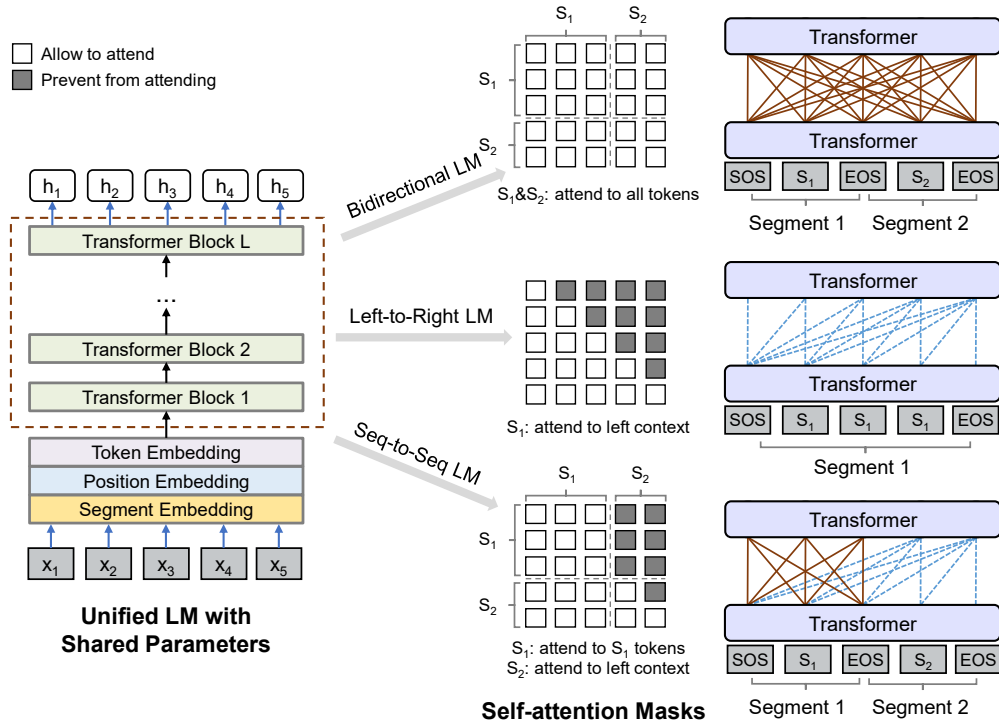

Figure 1: Overview of unified LM pre-training. The model parameters are shared across the LM objectives (i.e., bidirectional LM, unidirectional LM, and sequence-to-sequence LM). We use different self-attention masks to control the access to context for each word token. The right-to-left LM is similar to the left-to-right one, which is omitted in the figure for brevity.

representation. Once UNILM is pretrained, we can fine-tune it using task-specific data for downstream tasks.

## 2.1 Input Representation

The input $x$ is a word sequence, which is either a text segment for unidirectional LMs or a pair of segments packed together for bidirectional LM and sequence-to-sequence LM. We always add a special start-of-sequence ([SOS]) token at the beginning of input, and a special end-of-sequence ([EOS]) token at the end of each segment. [EOS] not only marks the sentence boundary in NLU tasks, but also is used for the model to learn when to terminate the decoding process in NLG tasks. The input representation follows that of BERT [9]. Texts are tokenized to subword units by WordPiece [47]. For each input token, its vector representation is computed by summing the corresponding token embedding, position embedding, and segment embedding. Since UNILM is trained using multiple LM tasks, segment embeddings also play a role of LM identifier in that we use different segment embeddings for different LM objectives.

## 2.2 Backbone Network: Multi-Layer Transformer

The input vectors $\{\mathbf{x}_i\}_{i=1}^{|x|}$ is first packed into $\mathbf{H}^0 = [\mathbf{x}_1, \cdots, \mathbf{x}_{|x|}]$, and then encoded into contextual representations at different levels of abstract $\mathbf{H}^l = [\mathbf{h}_1^l, \cdots, \mathbf{h}_{|x|}^l]$ using an $L$-layer Transformer $\mathbf{H}^l = \mathrm{Transformer}_l(\mathbf{H}^{l-1}), l \in [1, L]$. In each Transformer block, multiple self-attention heads are used to aggregate the output vectors of the previous layer. For the $l$-th Transformer layer, the output

of a self-attention head $\mathbf{A}_l$ is computed via:

$$\mathbf{Q} = \mathbf{H}^{l-1}\mathbf{W}_l^Q, \quad \mathbf{K} = \mathbf{H}^{l-1}\mathbf{W}_l^K, \quad \mathbf{V} = \mathbf{H}^{l-1}\mathbf{W}_l^V \tag{1}$$

$$\mathbf{M}_{ij} = \begin{cases} 0, & \text{allow to attend} \\ -\infty, & \text{prevent from attending} \end{cases} \tag{2}$$

$$\mathbf{A}_l = \text{softmax}(\frac{\mathbf{Q}\mathbf{K}^\intercal}{\sqrt{d_k}} + \mathbf{M})\mathbf{V}_l \tag{3}$$

where the previous layer's output $\mathbf{H}^{l-1} \in \mathbb{R}^{|x| \times d_h}$ is linearly projected to a triple of queries, keys and values using parameter matrices $\mathbf{W}_l^Q, \mathbf{W}_l^K, \mathbf{W}_l^V \in \mathbb{R}^{d_h \times d_k}$, respectively, and the mask matrix $\mathbf{M} \in \mathbb{R}^{|x| \times |x|}$ determines whether a pair of tokens can be attended to each other.

We use different mask matrices $\mathbf{M}$ to control what context a token can attend to when computing its contextualized representation, as illustrated in Figure 1. Take bidirectional LM as an example. The elements of the mask matrix are all 0s, indicating that all the tokens have access to each other.

## 2.3 Pre-training Objectives

We pretrain UNILM using four cloze tasks designed for different language modeling objectives. In a cloze task, we randomly choose some WordPiece tokens in the input, and replace them with special token [MASK]. Then, we feed their corresponding output vectors computed by the Transformer network into a softmax classifier to predict the masked token. The parameters of UNILM are learned to minimize the cross-entropy loss computed using the predicted tokens and the original tokens. It is worth noting that the use of cloze tasks makes it possible to use the same training procedure for all LMs, unidirectional and bidirectional alike.

**Unidirectional LM**　We use both left-to-right and right-to-left LM objectives. Take the left-to-right LM as an example. The representation of each token encodes only the leftward context tokens and itself. For instance, to predict the masked token of "$x_1 x_2$ [MASK] $x_4$", only tokens $x_1, x_2$ and itself can be used. This is done by using a triangular matrix for the self-attention mask $\mathbf{M}$ (as in Equation (2)), where the upper triangular part of the self-attention mask is set to $-\infty$, and the other elements to 0, as shown in Figure 1. Similarly, a right-to-left LM predicts a token conditioned on its future (right) context.

**Bidirectional LM**　Following [9], a bidirectional LM allows all tokens to attend to each other in prediction. It encodes contextual information from both directions, and can generate better contextual representations of text than its unidirectional counterpart. As indicated in Equation (2), the self-attention mask $\mathbf{M}$ is a zero matrix, so that every token is allowed to attend across all positions in the input sequence.

**Sequence-to-Sequence LM**　As shown in Figure 1, for prediction, the tokens in the first (source) segment can attend to each other from both directions within the segment, while the tokens of the second (target) segment can only attend to the leftward context in the target segment and itself, as well as all the tokens in the source segment. For example, given source segment $t_1 t_2$ and its target segment $t_3 t_4 t_5$, we feed input "[SOS] $t_1$ $t_2$ [EOS] $t_3$ $t_4$ $t_5$ [EOS]" into the model. While both $t_1$ and $t_2$ have access to the first four tokens, including [SOS] and [EOS], $t_4$ can only attend to the first six tokens.

Figure 1 shows the self-attention mask $\mathbf{M}$ used for the sequence-to-sequence LM objective. The left part of $\mathbf{M}$ is set to 0 so that all tokens can attend to the first segment. The upper right part is set to $-\infty$ to block attentions from the source segment to the target segment. Moreover, for the lower right part, we set its upper triangular part to $-\infty$, and the other elements to 0, which prevents tokens in the target segment from attending their future (right) positions.

During training, we randomly choose tokens in both segments, and replace them with the special token [MASK]. The model is learned to recover the masked tokens. Since the pair of source and target texts are packed as a contiguous input text sequence in training, we implicitly encourage the model to learn the relationship between the two segments. In order to better predict tokens in the target segment, UNILM learns to effectively encode the source segment. Thus, the cloze task designed for

the sequence-to-sequence LM, also known as the encoder-decoder model, simultaneously pre-trains a bidirectional encoder and an unidirectional decoder. The pre-trained model, used as an encoder-decoder model, can be easily adapted to a wide range of conditional text generation tasks, such as abstractive summarization.

**Next Sentence Prediction** For the bidirectional LM, we also include the next sentence prediction task for pre-training, as in [9].

## 2.4 Pre-training Setup

The overall training objective is the sum of different types of LM objectives described above. Specifically, within one training batch, $1/3$ of the time we use the bidirectional LM objective, $1/3$ of the time we employ the sequence-to-sequence LM objective, and both left-to-right and right-to-left LM objectives are sampled with rate of $1/6$. The model architecture of UNILM follows that of BERT$_{\text{LARGE}}$ [9] for a fair comparison. The gelu activation [18] is used as GPT [31]. Specifically, we use a 24-layer Transformer with $1,024$ hidden size, and 16 attention heads, which contains about 340M parameters. The weight matrix of the softmax classifier is tied with token embeddings. UNILM is initialized by BERT$_{\text{LARGE}}$, and then pre-trained using English Wikipedia[2] and BookCorpus [52], which have been processed in the same way as [9]. The vocabulary size is $28,996$. The maximum length of input sequence is $512$. The token masking probability is $15\%$. Among masked positions, $80\%$ of the time we replace the token with [MASK], $10\%$ of the time with a random token, and keeping the original token for the rest. In addition, $80\%$ of the time we randomly mask one token each time, and $20\%$ of the time we mask a bigram or a trigram.

Adam [22] with $\beta_1 = 0.9$, $\beta_2 = 0.999$ is used for optimization. The learning rate is 3e-5, with linear warmup over the first $40,000$ steps and linear decay. The dropout rate is $0.1$. The weight decay is $0.01$. The batch size is $330$. The pre-training procedure runs for about $770,000$ steps. It takes about 7 hours for $10,000$ steps using 8 Nvidia Telsa V100 32GB GPU cards with mixed precision training.

## 2.5 Fine-tuning on Downstream NLU and NLG Tasks

For NLU tasks, we fine-tune UNILM as a bidirectional Transformer encoder, like BERT. Take text classification as an example. We use the encoding vector of [SOS] as the representation of input, denoted as $\mathbf{h}_1^L$, and feed it to a randomly initialized softmax classifier (i.e., the task-specific output layer), where the class probabilities are computed as $\text{softmax}(\mathbf{h}_1^L \mathbf{W}^C)$, where $\mathbf{W}^C \in \mathbb{R}^{d_h \times C}$ is a parameter matrix, and $C$ the number of categories. We maximize the likelihood of the labeled training data by updating the parameters of the pre-trained LM and the added softmax classifier.

For NLG tasks, we take the sequence-to-sequence task as an example. The fine-tuning procedure is similar to pre-training using the self-attention masks as in Section 2.3. Let S1 and S2 denote source and target sequences, respectively. We pack them together with special tokens, to form the input "[SOS] S1 [EOS] S2 [EOS]". The model is fine-tuned by masking some percentage of tokens in the target sequence at random, and learning to recover the masked words. The training objective is to maximize the likelihood of masked tokens given context. It is worth noting that [EOS], which marks the end of the target sequence, can also be masked during fine-tuning, thus when this happens, the model learns when to emit [EOS] to terminate the generation process of the target sequence.

## 3 Experiments

We have conducted experiments on both NLU (i.e., the GLUE benchmark, and extractive question answering) and NLG tasks (i.e., abstractive summarization, question generation, generative question answering, and dialog response generation).

## 3.1 Abstractive Summarization

Automatic text summarization produces a concise and fluent summary conveying the key information in the input (e.g., a news article). We focus on abstractive summarization, a generation task where

| | RG-1 | RG-2 | RG-L |
|---|---|---|---|
| *Extractive Summarization* | | | |
| LEAD-3 | 40.42 | 17.62 | 36.67 |
| Best Extractive [27] | 43.25 | **20.24** | 39.63 |
| *Abstractive Summarization* | | | |
| PGNet [36] | 39.53 | 17.28 | 37.98 |
| Bottom-Up [16] | 41.22 | 18.68 | 38.34 |
| S2S-ELMo [13] | 41.56 | 18.94 | 38.47 |
| UNILM | **43.33** | 20.21 | **40.51** |

Table 3: Evaluation results on CNN/DailyMail summarization. Models in the first block are extractive systems listed here for reference, while the others are abstractive models. The results of the best reported extractive model are taken from [27]. RG is short for ROUGE.

| | RG-1 | RG-2 | RG-L |
|---|---|---|---|
| *10K Training Examples* | | | |
| Transformer [42] | 10.97 | 2.23 | 10.42 |
| MASS [38] | 25.03 | 9.48 | 23.48 |
| UNILM | **32.96** | **14.68** | **30.56** |
| *Full Training Set* | | | |
| OpenNMT [23] | 36.73 | 17.86 | 33.68 |
| Re3Sum [4] | 37.04 | 19.03 | 34.46 |
| MASS [38] | 37.66 | 18.53 | 34.89 |
| UNILM | **38.45** | **19.45** | **35.75** |

Table 4: Results on Gigaword abstractive summarization. Models in the first block only use 10K examples for training, while the others use 3.8M examples. Results of OpenNMT and Transformer are taken from [4, 38]. RG is short for ROUGE.

the summary is not constrained to reusing the phrases or sentences in the input text. We use the non-anonymized version of the CNN/DailyMail dataset [36] and Gigaword [35] for model fine-tuning and evaluation. We fine-tune UNILM as a sequence-to-sequence model following the procedure described in Section 2.5 by concatenating document (the first segment) and summary (the second segment) as input which is truncated according to a pre-defined maximum length.

We fine-tune our model on the training set for 30 epochs. We reuse most hyper-parameters from pre-training. The masking probability is $0.7$. We also use label smoothing [39] with rate of $0.1$. For CNN/DailyMail, we set batch size to $32$, and maximum length to $768$. For Gigaword, we set batch size to $64$, and maximum length to $256$. During decoding, we use beam search with beam size of $5$. The input document is truncated to the first $640$ and $192$ tokens for CNN/DailyMail and Gigaword, respectively. We remove duplicated trigrams in beam search, and tweak the maximum summary length on the development set [28, 13].

We use the F1 version of ROUGE [25] as the evaluation metric for both datasets. In Table 3, we compare UNILM against the baseline and several state-of-the-art models on CNN/DailyMail. LEAD-3 is a baseline model that extracts the first three sentences in a document as its summary. PGNet [36] is a sequence-to-sequence model based on the pointer-generator network. S2S-ELMo [13] uses a sequence-to-sequence model augmented with pre-trained ELMo representations, which is termed as SRC-ELMO+SHDEMB in [13]. Bottom-Up [16] is a sequence-to-sequence model augmented with a bottom-up content selector for selecting salient phrases. We also include in Table 3 the best reported extractive summarization result [27] on the dataset. As shown in Table 3, our model outperforms all previous abstractive systems, creating a new state-of-the-art abstractive summarization result on the dataset. Our model also outperforms the best extractive model [27] by $0.88$ point in ROUGE-L.

In Table 4, we evaluate the models on Gigaword with different scales (10K and 3.8M). Both Transformer [42] and OpenNMT [23] implement standard attentional sequence-to-sequence models. Re3Sum [4] retrieves summaries as candidate templates, and then use an extended sequence-to-sequence model to generate summaries. MASS [38] is a pre-trained sequence-to-sequence model based on Transformer networks. Experimental results show that UNILM achieves better performance than previous work. Besides, in the low-resource setting (i.e., only 10,000 examples are used as training data), our model outperforms MASS by $7.08$ point in ROUGE-L.

## 3.2 Question Answering (QA)

The task is to answer a question given a passage [32, 33, 15]. There are two settings. The first is called *extractive* QA, where the answer is assumed to be a text span in the passage. The other is called *generative* QA, where the answer needs to be generated on the fly.

**Extractive QA** This task can be formulated as a NLU task where we need to predict the start and end positions of the answer spans within the passage. We fine-tune the pre-trained UNILM as a

| | EM | F1 |
|---|---|---|
| RMR+ELMo [20] | 71.4 | 73.7 |
| BERT$_{\text{LARGE}}$ | 78.9 | 81.8 |
| UNILM | **80.5** | **83.4** |

Table 5: Extractive QA results on the SQuAD development set.

| | F1 |
|---|---|
| DrQA+ELMo [34] | 67.2 |
| BERT$_{\text{LARGE}}$ | 82.7 |
| UNILM | **84.9** |

Table 6: Extractive QA results on the CoQA development set.

| | F1 |
|---|---|
| Seq2Seq [34] | 27.5 |
| PGNet [34] | 45.4 |
| UNILM | **82.5** |

Table 7: Generative QA results on the CoQA development set.

bidirectional encoder for the task. We conduct experiments on the Stanford Question Answering Dataset (SQuAD) 2.0 [33], and Conversational Question Answering (CoQA) [34] datasets.

The results on SQuAD 2.0 are reported in Table 5, where we compare two models in Exact Match (EM) and F1 score. RMR+ELMo [20] is an LSTM-based question answering model augmented with pre-trained language representation. BERT$_{\text{LARGE}}$ is a cased model, fine-tuned on the SQuAD training data for 3 epochs, with batch size 24, and maximum length 384. UNILM is fine-tuned in the same way as BERT$_{\text{LARGE}}$. We see that UNILM outperforms BERT$_{\text{LARGE}}$.

CoQA is a conversational question answering dataset. Compared with SQuAD, CoQA has several unique characteristics. First, the examples in CoQA are conversational, so we need to answer the input question based on conversation histories. Second, the answers in CoQA can be free-form texts, including a large portion is of yes/no answers.

We modify the model used for SQuAD as follows. Firstly, in addition to the asked question, we concatenate the question-answer histories to the first segment, so that the model can capture conversational information. Secondly, for yes/no questions, we use the final hidden vector of the `[SOS]` token to predict whether the input is a yes/no question, and whether the answer is `yes` or `no`. For other examples, we select a passage subspan with the highest F1 score for training.

The results on CoQA are reported in Table 6, where we compare two models in F1 scores. DrQA+ELMo [34] is an LSTM-based question answering model augmented with pre-trained ELMo representation. BERT$_{\text{LARGE}}$ is a cased model, fine-tuned on the CoQA training data for 2 epochs, with batch size 16, and maximum length 512. UNILM is fine-tuned with the same hyper-parameters as BERT$_{\text{LARGE}}$. We see that UNILM outperforms BERT$_{\text{LARGE}}$.

**Generative QA** Generative question answering generates free-form answers for the input question and passage, which is a NLG task. In contrast, extractive methods can only predict subspans of the input passage as answers. On the CoQA dataset (as described above), Reddy et al. [2019] show that vanilla sequence-to-sequence models still underperforms extractive methods by a wide margin.

We adapt UNILM to generative question answering as a sequence-to-sequence model. The first segment (i.e., the input sequence) is the concatenation of conversational histories, the input question and the passage. The second segment (i.e., the output sequence) is the answer. We fine-tune the pre-trained UNILM on the CoQA training set for 10 epochs. We set the batch size to 32, the mask probability to 0.5, and the maximum length to 512. We also use label smoothing with rate of 0.1. The other hyper-parameters are kept the same as pre-training. During decoding, we use beam search with beam size of 3. The maximum length of input question and passage is 470. For passages that are longer than the maximum length, we split the passage into several chunks with a sliding window approach, and select a chunk with the highest word overlap over the question.

We compare our method with the generative question answering models Seq2Seq and PGNet as described in [34]. The Seq2Seq baseline is a sequence-to-sequence model with an attention mechanism. The PGNet model augments Seq2Seq with a copy mechanism. As shown in Table 7, our generative question answering model outperforms previous generative methods by a wide margin, which significantly closes the gap between generative method and extractive method.

### 3.3 Question Generation

We conduct experiments for the answer-aware question generation task [51]. Given an input passage and an answer span, our goal is to generate a question that asks for the answer. The SQuAD 1.1 dataset [32] is used for evaluation. Following [12], we split the original training set into training and

| | BLEU-4 | MTR | RG-L |
|---|---|---|---|
| CorefNQG [11] | 15.16 | 19.12 | - |
| SemQG [49] | 18.37 | 22.65 | 46.68 |
| UNILM | **22.12** | **25.06** | **51.07** |
| MP-GSN [50] | 16.38 | 20.25 | 44.48 |
| SemQG [49] | 20.76 | 24.20 | 48.91 |
| UNILM | **23.75** | **25.61** | **52.04** |

Table 8: Question generation results on SQuAD. MTR is short for METEOR, and RG for ROUGE. Results in the groups use different data splits.

| | EM | F1 |
|---|---|---|
| UNILM QA Model (Section 3.2) | 80.5 | 83.4 |
| + UNILM Generated Questions | **84.7** | **87.6** |

Table 9: Question generation based on UNILM improves question answering results on the SQuAD development set.

| | NIST-4 | BLEU-4 | METEOR | Entropy-4 | Div-1 | Div-2 | Avg len |
|---|---|---|---|---|---|---|---|
| Best System in DSTC7 Shared Task | 2.523 | 1.83 | 8.07 | 9.030 | 0.109 | 0.325 | 15.133 |
| UNILM | **2.669** | **4.39** | **8.27** | **9.195** | **0.120** | **0.391** | 14.807 |
| Human Performance | 2.650 | 3.13 | 8.31 | 10.445 | 0.167 | 0.670 | 18.76 |

Table 10: Response generation results. Div-1 and Div-2 indicate diversity of unigrams and bigrams, respectively.

test sets, and keep the original development set. We also conduct experiments following the data split as in [50], which uses the reversed dev-test split.

The question generation task is formulated as a sequence-to-sequence problem. The first segment is the concatenation of input passage and answer, while the second segment is the generated question.

We fine-tune UNILM on the training set for 10 epochs. We set batch size to 32, masking probability to 0.7, and learning rate to 2e-5. The rate of label smoothing is 0.1. The other hyper-parameters are the same as pre-training. During decoding, we truncate the input to 464 tokens by selecting a passage chunk which contains the answer. The evaluation metrics BLEU-4, METEOR, and ROUGE-L are computed by the same scripts as in [12].

The results[3] are presented in Table 8. CorefNQG [11] is based on a sequence-to-sequence model with attention and a feature-rich encoder. MP-GSN [50] uses an attention-based sequence-to-sequence model with a gated self-attention encoder. SemQG [49] uses two semantics-enhanced rewards to regularize the generation. UNILM outperforms previous models and achieves a new state-of-the-art for question generation.

**Generated Questions Improve QA**    The question generation model can automatically harvest a large number of question-passage-answer examples from a text corpus. We show that the augmented data generated by question generation improves the question answering model.

We generate five million answerable examples, and four million unanswerable examples by modifying the answerable ones. We fine-tune our question answering model on the generated data for one epoch. Then the model is fine-tuned on the SQuAD 2.0 data for two more epochs.

As shown in Table 9, the augmented data generated by UNILM improves question answering model introduced in Section 3.2. Note that we use bidirectional masked language modeling as an auxiliary task for both the generated and SQuAD 2.0 datasets during fine-tuning, which brings 2.3 absolute improvement compared to directly using automatically generated examples. A possible reason is that the auxiliary task alleviates catastrophic forgetting [48] when fine-tuning on augmented data.

## 3.4   Response Generation

We evaluate UNILM on the document-grounded dialog response generation task [30, 15]. Given a multi-turn conversation history and a web document as the knowledge source, the system needs to

| Model | CoLA MCC | SST-2 Acc | MRPC F1 | STS-B S Corr | QQP F1 | MNLI-m/mm Acc | QNLI Acc | RTE Acc | WNLI Acc | AX Acc | **Score** |
|---|---|---|---|---|---|---|---|---|---|---|---|
| GPT | 45.4 | 91.3 | 82.3 | 80.0 | 70.3 | 82.1/81.4 | 87.4 | 56.0 | 53.4 | 29.8 | 72.8 |
| BERT$_{LARGE}$ | 60.5 | **94.9** | 89.3 | 86.5 | **72.1** | 86.7/**85.9** | **92.7** | 70.1 | 65.1 | 39.6 | 80.5 |
| UNILM | **61.1** | 94.5 | **90.0** | **87.7** | 71.7 | **87.0/85.9** | **92.7** | **70.9** | 65.1 | 38.4 | **80.8** |

Table 11: GLUE test set results scored using the GLUE evaluation server.

generate a natural language response that is both conversationally appropriate and reflective of the contents of the web document. We fine-tune UNILM to the task as a sequence-to-sequence model. The first segment (input sequence) is the concatenation of the web document and the conversation history. The second segment (output sequence) is the response. We fine-tune UNILM on the DSTC7 training data for 20 epochs, with batch size 64. The masking probability is set to 0.5. The maximum length is 512. During decoding, we use beam search with size of 10. The maximum length of generated response is set to 40. As shown in Table 10, UNILM outperforms the best system [40] in the DSTC7 shared task [14] across all evaluation metrics.

## 3.5 GLUE Benchmark

We evaluate UNILM on the General Language Understanding Evaluation (GLUE) benchmark [44]. GLUE is a collection of nine language understanding tasks, including question answering [32], linguistic acceptability [45], sentiment analysis [37], text similarity [5], paraphrase detection [10], and natural language inference (NLI) [7, 2, 17, 3, 24, 46].

Our model is fine-tuned as a bidirectional LM. We use Adamax [21] as our optimizer with a learning rate of 5e-5 and a batch size of 32. The maximum number of epochs is set to 5. A linear learning rate decay schedule with warmup of 0.1 is used. The dropout rate of the last linear projection for each task is set to 0.1, except 0.3 for MNLI and 0.05 for CoLA/SST-2. To avoid the gradient explosion issue, the gradient norm was clipped within 1. We truncated the tokens no longer than 512.

Table 11 presents the GLUE test results obtained from the benchmark evaluation server. The results show that UNILM obtains comparable performance on the GLUE tasks in comparison with BERT$_{LARGE}$.

## 4 Conclusion and Future Work

We propose a unified pre-training model, UNILM, which is jointly optimized for several LM objectives with shared parameters. The unification of bidirectional, unidirectional, and sequence-to-sequence LMs enables us to straightforwardly fine-tune the pre-trained UNILM for both NLU and NLG tasks. Experimental results demonstrate that our model compares favorably with BERT on the GLUE benchmark and two question answering datasets. In addition, UNILM outperforms previous state-of-the-art models on five NLG datasets: CNN/DailyMail and Gigaword abstractive summarization, SQuAD question generation, CoQA generative question answering, and DSTC7 dialog response generation.

The work can be advanced from the following perspectives:

- We will push the limit of the current method by training more epochs and larger models on web-scale text corpora. At the same time, we will also conduct more experiments on end applications as well as ablation experiments to investigate the model capability and the benefits of pre-training multiple language modeling tasks with the same network.
- We are focusing on monolingual NLP tasks in our current experiments. We are also interested in extending UNILM to support cross-lingual tasks [6].
- We will conduct multi-task fine-tuning on both NLU and NLG tasks, which is a natural extension of Multi-Task Deep Neural Network (MT-DNN) [26].

**Acknowledgement**   We would like to acknowledge Shiyue Zhang for the helpful discussions about the question generation experiments.

## Footnotes

[2]Wikipedia version: enwiki-20181101.

[3]Notice that if we directly use the tokenized references provided by Du et al. [2017], the results are (21.63 BLEU-4 / 25.04 METEOR / 51.09 ROUGE-L) on the original data split [12], and (23.08 BLEU-4 / 25.57 METEOR / 52.03 ROUGE-L) in the reversed dev-test setup [50].

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
