[Supplementary Material]

# Supplementary Material:
# Unified Language Model Pre-training for
# Natural Language Understanding and Generation

**Li Dong**[*]   **Nan Yang**[*]   **Wenhui Wang**[*]   **Furu Wei**[*†]   **Xiaodong Liu**   **Yu Wang**
**Jianfeng Gao**   **Ming Zhou**   **Hsiao-Wuen Hon**
Microsoft Research
{lidong1,nanya,wenwan,fuwei}@microsoft.com
{xiaodl,yuwan,jfgao,mingzhou,hon}@microsoft.com

## 1   Long Text Generation: A Case Study

Our model can generate text samples using the left-to-right setting. We picked three text samples sampled from left to right using our model, as shown in Table 1. We use the top-$40$ truncating sampling strategy [1], and forbid duplicate 4-grams during generation. For each example, we sampled 10 times from the same input and we hand-picked the best one; as such, these samples should be considered to be better than the average model output. From the examples, we find that the model can produce fluent output with somewhat consistent contents which fits the inputs' genres and topics. In the first example, given a modified excerpt from the novel "*1984*" as input, the model's output is like a short paragraph in an fantasy novel; for the other two with input from Wikipedia, the model makes up "facts" with plausible names, time stamps and events written in the Wikipedia style.

## 2   GLUE Benchmark

As shown in Table 2, we summarize the data size and the evaluation metrics used for the General Language Understanding Evaluation (GLUE) benchmark.

## References

[1] Alec Radford, Jeff Wu, Rewon Child, David Luan, Dario Amodei, and Ilya Sutskever. Language models are unsupervised multitask learners. 2019.

---

[*] Equal contribution. † Contact person.

| Input | Output |
|---|---|
| Winston sat back. A sense of complete helplessness had descended upon him. | *Winston sat back. A sense of complete helplessness had descended upon him.* He was now helpless and unmoving. What would that mean? To become a monk under her mother's rule. What would it mean for him to be put at the center of the most dangerous of all of demons, the demon known as Azrael? He would never know exactly what that was, or even how it would end, unless he could sense the strength of Azrael's mind and soul. It wasn't only the existence of his mother, it was all he knew, and every moment. He was trapped within her embrace, a trapped, torn, battered, crippling presence he only knew it would mean was eternity. And in the instant that he was lost his life changed forever, when Azraelis' spirit touched his mind, he was engulfed in this dark and mystical, so enchanting and beautiful force. (...) |
| Bert is a yellow Muppet character on the long running children's television show, Sesame Street. | *Bert is a yellow Muppet character on the long running children's television show, Sesame Street.* He would typically appear as a sidekick or as a villain in the show; he is usually portrayed as a cowardly and untrustworthy figure that tends to intrude on the normal life of the street, such as "The Story of the Mole" , "The Story from the Side", and "Bump the Bush". His voice was provided by Michael Combs. Bert was a regular cast member of Sesame Street, and his appearance was always played throughout the series by two of his regular cast members, Frank Laughner and Michael Combs. (...) |
| Maurice Ravel (7 March 1875 - 28 December 1937) was a French composer, pianist and conductor. | *Maurice Ravel (7 March 1875 - 28 December 1937) was a French composer, pianist and conductor.* He was born in Paris, and studied piano with Eugene Massenet. In the early 1900s Ravel became interested in the composition of music, and he composed much of his early music for the piano. Ravel's early work is marked by a marked sense of lyricism, but in the later years of his career he developed a strong preference for orchestral forms. His first work, "le Grand Cours d' Auvergne", was composed in 1909 by Maurice Ravel, a student of Jules Massenet, and was published by A.F.A. in 1912. It was re-published in 1912 by the publisher J.S.D.M. de l'Etablissement Musicale de la Musique Francaise. Ravel wrote the piano concerto "la Tragedie et la Chanson Dans le Theatre des Champs Elysees" in 1916. (...) |

Table 1: Text samples generated by our model using left-to-right generation.

| Corpus | #Train/#Dev/#Test | Metrics |
|---|---|---|
| *Single-Sentence Classification* | | |
| CoLA (Acceptability) | 8.5k/1k/1k | Matthews corr |
| SST-2 (Sentiment) | 67k/872/1.8k | Accuracy |
| *Pairwise Text Classification* | | |
| MNLI (NLI) | 393k/20k/20k | Accuracy |
| RTE (NLI) | 2.5k/276/3k | Accuracy |
| QNLI (NLI) | 108k/5.7k/5.7k | Accuracy |
| WNLI (NLI) | 634/71/146 | Accuracy |
| QQP (Paraphrase) | 364k/40k/391k | F1 score |
| MRPC (Paraphrase) | 3.7k/408/1.7k | F1 score |
| *Text Similarity* | | |
| STS-B (Similarity) | 7k/1.5k/1.4k | Spearman corr |

Table 2: Summary of the GLUE benchmark.