[Reviews · NeurIPS 2019]

Reviewer 1



This paper provides a method to pretrain a single Transformer architecture on three objectives: (i) unidirectional language model (e.g. like GPT-2), (ii) bidirectional language model (e.g. like BERT), and (iii) sequence-to-sequence language model (e.g. like a standard encoder-decoder architecture, with a bidirectional network to encode the source information, paired with a unidirectional decoder to generate the target sequence). This unified architecture circumvents the shortcoming of both models like BERT (which can condition on bidirectional context, but harder to use for downstream tasks that involve generation due to bidirectionality) and GPT-2 (easy to apply for generation tasks since it works left-to-right, but bidirectional encoders have been known to work much better than unidirectional ones in sequence-to-sequence models), and thereby combines the best of both worlds. This is done using a simple masking scheme that restricts which words the model can pay attention to, depending on which objective function is used (e.g. if using a unidirectional, left-to-right objective, then all tokens to the right of the target word are masked out). Experiments on text summarisation (CNN/DailyMail and Gigaword), question answering (SQuAD, CoQA extractive, and CoQA abstractive), question generation, and GLUE indicate that the proposed pretraining approach largely matches or surpasses the current state of the art. Originality: this paper addresses an important problem of unifying the different language modelling pretraining objectives (unidirectional/bidirectional/seq2seq) with a single model, thus circumventing the limitations of earlier work. Their masking approach crucially enables pretraining the two key ingredients of sequence-to-sequence models with a single model: (i) a bidirectional encoder, and (ii) a unidirectional decoder. In earlier work, the advantage of language model pretraining has mostly been shown for classification tasks, but this work crucially demonstrates a way of incorporating such pretraining method for language generation tasks (with substantial improvements to show for summarisation, question answering, and question generation). - Quality: The proposed masking approach is simple yet effective. All the claims of the method's success are backed by the empirical findings on multiple tasks and datasets. - Clarity: This paper clearly explains the method used, such as using Figure 1. Overall the method is presented well and the paper is easy-to-understand. The paper also contains sufficient technical details, such as the hyper-parameters used and any additional tweaks, in order to help reproducibility. I have some minor comments regarding the presentation in point 5 ("Improvements") below. - Significance: This paper introduces an effective way to combine different language modelling objectives for improving text generation in sequence-to-sequence models, which is an important and welcome contribution in the emerging field of transfer learning in NLP. ----- Update after authors' response: the response addressed most of my concerns. Overall, I believe that the method and empirical findings help make progress towards data-efficient NLP (particularly for generation tasks), and would be useful for the community. I am therefore maintaining my overall score of "7".

Reviewer 2



1, There do exist challenges applying pretraining LMs to both NLU and NLG tasks. This paper takes advantage of BERT’s masked LM objective and GPT’s architecture. Combining three types of LM objectives is a straightforward but effective extension. 2, For the second advantage of this paper (line 45-48), there is no experiments to compare pretrained LMs with different objectives. I also have doubts on why single objective LMs will overfit since it is trained on large scale corpus. 3, The generation experiments are not convincing enough because the authors only conduct experiments on summarization and question generation. I would like to see more experimental results on other generation tasks such as machine translation and response generation.

Reviewer 3



I have read the authors' response. While the response doesn't fully answer my concern, i.e., providing results of training their large-sized model from scratch, it is a reasonable response. Also given the results and experiments conducted, I think the paper benefits the community. As such I have increased the score from 6 to 7. --- This paper extends BERT to propose a multi-task pretraining approach. The tasks are language modeling (left-to-right, right-to-left), BERT's objective, and seq2seq generation. All the three tasks share the same Transformer backbone (BERT large) and different task are accomplished through masking matrices. The authors presented strong improvements across a wide range of tasks, such as abstractive summarization (CNN/DailyMail), question answering (SQuAD, CoQA) using both extractive and generative approaches, and question generation. My only concern point is the initialization of this model from BERT large: what happens if this model is trained from scratch? or what happens if BERT large is continued to be trained on the version of their data?

[Author Response · NeurIPS 2019]

# 1 Response to Reviewer #1

**Interesting to see how well the proposed model would do under such zero-shot setup (i.e. without fine-tuning the model on any particular supervised task).** We compared with GPT-2 (345M) on the Winograd Schema Challenge dataset under the zero-shot setup following the GPT-2 paper. Although GPT-2 is trained on much larger corpus, UNILM can achieve slightly better accuracy with comparable number of model parameters.

| Model | Number of Parameters | Pretraining Data Size | Accuracy (%) |
|---|---|---|---|
| GPT-2 (345M) | 345 million | 40GB | 62.25 |
| UNILM | 340 million | 15GB | **64.47** |

Table 1: Results on Winograd Schema Challenge under zero-shot setup. GPT-2 accuracy is taken from their paper.

**Explain a bit more on what dataset the model was pretrained on, how this dataset was selected, and how the size of the pretraining dataset compares with e.g. ELMo or BERT.** We followed the same protocol of pretraining data as BERT. The BERT paper reports that BooksCorpus and Wikipedia contain 0.8B and 2.5B words, respectively. For our processed data, BooksCorpus and Wikipedia contain 0.75B and 2B words, respectively. ELMo was pretrained on Billion Word Benchmark (Chelba et al., 2014), which contains 0.83B tokens.

**Explain a bit more on the segment embedding.** The implementation is the same as word embedding, i.e., a lookup table is used to store the embeddings of segment indices. We assign a learnable embedding for each segment (such as "Segment 1", and "Segment 2") and feed it to model input, which indicates the segment of input tokens.

**Mention pretraining time in L150.** Thanks for the suggestion. We will update it in the revised version of the paper.

**Interesting to see what happens if beam search decoding is replaced with top-k sampling or nucleus sampling?** Top-k sampling and nucleus sampling improve the diversity of unconditioned generation (i.e., sampling text from language models). For conditioned generation (such as summarization, and question generation), beam search achieves better performance in terms of automatic evaluation metrics.

# 2 Response to Reviewer #2

**For the second advantage (L45-48), why single objective LMs will overfit since it is trained on large scale corpus?** Using one LM objective makes pre-training biased to a single type of attention pattern. For example, left-to-right LM pretrains how to attend the left context, but the encoders of seq-to-seq downstream tasks need to learn how to utilize both left and right context, which can not be pretrained by only using left-to-right LM objective. We will reword the sentence to avoid confusion.

**More experimental results on other generation tasks such as machine translation and response generation.** We also evaluate UNILM on a document grounded response generation dataset ("*[ACL-19] Conversing by Reading: Contentful Neural Conversation with On-demand Machine Reading*"). As shown in Table 2, UNILM[1] outperforms the best system (i.e., Team B) in the DSTC7 shared task "*End-to-End Conversation Modeling: Moving beyond Chitchat*".

| | NIST-4 | BLEU-4 | METEOR | Entropy-4 | Div-1 | Div-2 | Avg len |
|---|---|---|---|---|---|---|---|
| Best System (Team B) in DSTC7 Shared Task | 2.523 | 1.83 | 8.07 | 9.030 | 0.109 | 0.325 | 15.133 |
| UNILM | **2.669** | **4.39** | **8.27** | **9.195** | **0.120** | **0.391** | 14.807 |
| Human Performance | 2.650 | 3.13 | 8.31 | 10.445 | 0.167 | 0.670 | 18.76 |

Table 2: Response generation results.

# 3 Response to Reviewer #3

**My only concern point is the initialization of this model from BERT large: what happens if this model is trained from scratch? or what happens if BERT large is continued to be trained on the version of their data?** Under the setting of a smaller model size (i.e., BERT-base), we tried both training from scratch and initializing from BERT-base. Both initialization methods on downstream tasks can achieve similar performance, but initializing from BERT-base reduces the number of learning steps. In order to shorten the training time of our large-size model, we initialize it from BERT-large. We will also release a model trained from scratch. We further trained BERT-large using the same hyper-parameters, but the resulted model didn't significantly improve downstream tasks compared to original BERT-large. However, recent work[2] from Facebook (RoBERTa) shows that carefully tuning hyper-parameters, using more training data, and longer training time can improve BERT performance on several language understanding tasks. It is definitely worth further studying how our model will perform with a thorough hyper-parameter tuning and more training data.

## Footnotes

[1]Fine-tuning as a sequence-to-sequence model: 20 epochs; batch size=64; masking probability=0.5; maximum length=512. Decoding: beam search with beam size=10; maximum response length=40.

[2]https://arxiv.org/abs/1907.11692


[Meta-Review · NeurIPS 2019]

This paper presents an alternative training regime for the BERT contextual embedding model that incorporates additional conditioning contexts such as left to right language modelling and sequence transduction. The reviewers agree that the work is well motivated and is a reasonable attempt to address some of the issues with the original BERT model. The results are suitably strong, and as such this paper is likely to be of interest to those working on contextual embedding models, although it is puzzling that a classic language modelling perplexity evaluation was not included, given this is one of the objectives that the model optimises. The author's final paper should incorporate the answers to the questions raised by the reviewers.